# Improving Breast Cancer Outcomes for Indigenous Women in Australia

**DOI:** 10.3390/cancers16091736

**Published:** 2024-04-29

**Authors:** Vita Christie, Lynette Riley, Deb Green, Janaki Amin, John Skinner, Chris Pyke, Kylie Gwynne

**Affiliations:** 1Djurali Centre for Aboriginal and Torres Strait Islander Research and Education, Heart Research Institute, Sydney, NSW 2042, Australia; john.skinner@hri.org.au (J.S.); kylie.gwynne@hri.org.au (K.G.); 2Sydney School of Education & Social Work, The University of Sydney, Camperdown, NSW 2006, Australia; lynette.riley@sydney.edu.au; 3Armajun Aboriginal Health Service, Armidale, NSW 2350, Australia; dgreen@armajun.org.au; 4Department of Health Sciences, Macquarie University, Sydney, NSW 2109, Australia; janaki.amin@mq.edu.au; 5Royal Australasian College of Surgeons, Melbourne, VIC 3002, Australia; christophermpyke@outlook.com.au

**Keywords:** breast cancer, Indigenous health, Aboriginal and Torres Strait Islander, health policy, cancer health service delivery

## Abstract

**Simple Summary:**

The current evidence regarding Indigenous* women and breast cancer in Australia shows lower prevalence but higher mortality rates. There are a range of reasons for this, including co-morbidities, lack of access to health services and low health information fluency. Perhaps most importantly, breast cancer health policy and service delivery practice do not meet the needs of Indigenous women in Australia, according to Indigenous women. Talking and listening to Indigenous women about breast cancer highlight that the solutions to improve breast cancer outcomes are available and that they are not complex. Indigenous women must be involved in the improvement of policy and practice in order for these outcomes to improve. *Terminology: We respectfully refer to Aboriginal and Torres Strait Islander people as “Indigenous”.

**Abstract:**

In Australia, the incidence rate of breast cancer is lower in Indigenous* women than non-Indigenous women; however, the mortality rate is higher, with Indigenous women 1.2 times more likely to die from the disease. This paper provides practical and achievable solutions to improve health outcomes for Indigenous women with breast cancer in Australia. This research employed the Context–Mechanism–Outcome (CMO) framework to reveal potential mechanisms and contextual factors that influence breast cancer outcomes for Indigenous women, stratified into multiple levels, namely, micro (interpersonal), meso (systemic) and macro (policy) levels. The CMO framework allowed us to interpret evidence regarding Indigenous women and breast cancer and provides nine practical ways to improve health outcomes and survival rates.

## 1. Introduction

### 1.1. Breast Cancer in Indigenous Women in Australia

In Australia, breast cancer is the most diagnosed cancer and the second largest cause of cancer death, in women. Despite a lower prevalence of breast cancer in Indigenous women than non-Indigenous women [1], the mortality rate is higher; Indigenous women in Australia are 1.2 times more likely to die from the disease [2]. In New South Wales alone, Indigenous women are 69% more likely to die from breast cancer when compared to non-Indigenous women [3]. There are numerous factors that contribute to these statistics, including lower participation in screening services, socioeconomic disadvantage, younger age at diagnosis, geographic remoteness, co-morbidities and a more advanced stage of cancer at the time of diagnosis [1,3,4].

### 1.2. Context

To understand why Aboriginal and Torres Strait Islander peoples in Australia experience significantly poorer health outcomes when compared to the non-Indigenous population, it is important to recognise and acknowledge the devastating and ongoing impact of colonisation. The violent policies of elimination and control, the introduction of foreign diseases and the disenfranchisement of the Indigenous population from their land, family and community destroyed the well-cultivated balance that Indigenous people had with the land and nature for at least 60,000 years prior to colonisation. The introduction of rations, alcohol, refined foods or indeed the withholding of food altogether, took an enormous toll on the population, physically, mentally and spiritually. Colonisation denied Indigenous people full citizenship rights and access to healthcare and self-determination, and the effects continue today with ongoing racism, trauma and intergenerational trauma [5,6,7,8,9,10]. These effects have all contributed to what is currently referred to as a significant gap in life expectancy and poorer health outcomes for Indigenous people in Australia. This situation, combined with health services built on western models of care, has culminated in a situation where there are not enough health services offering culturally safe and welcoming services for a population that experiences the highest need in Australia.

The roles of social and, more importantly, cultural determinants of health for Indigenous Australians are indisputably substantiated [11,12,13]. The widespread effects of colonisation include, but are not limited to, enforced separation and/or disconnection from family and culture; food and resource insecurity and often poverty; and systemic, institutional and individual racism. The effects of these have culminated in poorer physical and mental health and more prevalent chronic health conditions and co-morbidities [14,15].

Cultural safety and appropriate practice in health service provision have been acknowledged widely as the keys to increasing the utilisation of health services by Indigenous people in Australia [13,16,17] and internationally [18,19]. In Australia, Aboriginal Community-controlled health organisations (ACCHOs) have been identified as being imperative to make substantial improvements in health outcomes. However, it needs to be acknowledged that much of the care of Indigenous people takes place outside of ACCHOs, and therefore, all healthcare services need to be culturally safe in order for health outcomes to improve [17,20,21].The approach whereby culturally appropriate and community-based services are incorporated into design has enabled improvements and increased the value and effectiveness across various areas of Indigenous health [22,23].

Aboriginal Community Controlled Health Organisations (ACCHOs) currently have a marginal role in breast cancer care, and mainstream services have made little effort to customise their services to culture and context. Combined, this is likely a factor in low screening rates and far poorer breast cancer outcomes for Indigenous women. There is some evidence [24,25,26,27,28] that cultural adaptation of breast cancer screening programmes has improved participation rates for Indigenous women around the world, but little evidence has been produced in Australia thus far.

### 1.3. Research into Breast Cancer Policy and Practice for Indigenous Women in Australia

While there is a breadth of research that interrogates the factors that affect Indigenous health in general, there is limited evidence that looks specifically at breast cancer policy and practice in Australia for Indigenous women [1,4,29,30]. More recently, this author team has taken a deeper look at where policy and practice are not meeting the needs of Indigenous women in Australia [31,32,33]. By critically evaluating evidence and exploring the implications for practice, this paper aims to bring the learnings together and apply a framework to understand the areas in which change needs to, and can, feasibly occur.

### 1.4. Health Information Fluency

The term “health literacy” has traditionally been used to describe an individual’s ability to find and interpret health information. This term has more recently been brought into question in relation to the implied onus on the individual rather than the role of the health information provider(s). It has therefore been proposed that a new term be introduced, “health information fluency”, to encompass an all-inclusive and effective use of health information [34]. This translates to health service providers using plain English and having a shared vocabulary when communicating with Indigenous women in order to enhance understanding. This will likely benefit every patient.

As shown by Dr. Lynette Riley in Chapter One of the book entitled Community Led Research [35], it is not only incumbent on Indigenous women to increase literacy around health but also imperative that service providers are familiar with and understand the cultural needs of the women. As shown in Figure 1, when the Indigenous community and the service provider are fully aware of one another and are engaged in open and regular communication, it will lead to growth and sustainable change for all involved.

### 1.5. Study Novelty

The novelty of this study is bringing together the evidence regarding practice and policy and privileging the voices of Indigenous women throughout. When we look at the evidence regarding health outcomes and compare this to the policy that drives the practice, there is a deep divide. This study gathers this perspective and provides practical solutions to remedy the issues.

## 2. Methods

### 2.1. Study Design

This research employed the Context–Mechanism–Outcome (CMO) framework. The CMO framework is an implementation science tool used to reveal potential mechanisms and contextual factors that influence the outcomes of health programmes and health policies in the real world. It is used to explain generative causation, which helps to highlight the relationship between the context, mechanism, and outcome of a program or policy [36].

### 2.2. Procedures and Assessment

This framework has been applied to the results of the research in order to inform recommended changes. The mechanisms and contextual factors have been stratified into multiple levels: micro (interpersonal), meso (systemic) and macro (policy) levels. It used “If-Then-Because” statements to broadly represent the elements of the CMO framework.

### 2.3. Protocol Selection

The research, which had the CMO framework applied to it, comprised a systematic review of the evidence about culturally safe care improving outcomes for Indigenous women with breast cancer around the world, a protocol for improving breast cancer outcomes for Aboriginal women in Australia, qualitative research asking Aboriginal women about what they think will improve breast cancer outcomes and a review of the breast cancer policy in Australia, with recommendations made for how it might better meet the needs of Aboriginal and Torres Strait Islander women.

## 3. Results

This study identified nine recommendations to improve breast cancer outcomes for Indigenous women. These recommendations were developed from the CMO analysis, as shown in Table 1 and Table 2. Table 1 and Table 2 were developed to clearly develop likely scenarios, both from the perspective of making some changes and the perspective of not making the changes.

The CMO process identified nine practical ways to improve health outcomes and survival rates for Indigenous women in Australia:Improved health information fluency regarding the importance of screening and early detection and ensuring this is delivered in a culturally safe way.Increased cultural safety and access to screening services.Presence of Indigenous health service providers at screening and follow-up supportEnsuring the health service provider is local and/or familiar with Indigenous culture and community.Culturally safe screening, diagnosis, treatment and follow-up settings.Local Indigenous screening champions in the community.Protocols for abnormal results are culturally safe and tailored to the local setting and community.Culturally safe support is available for people who receive diagnoses.The policy is re-written in consultation with Indigenous people and provides solutions to improve health outcomes.

The process and impact of enablers and barriers to improved breast cancer outcomes for Indigenous women are shown in Figure 2. This figure similarly divides both the barriers and enablers into three separate levels, namely, micro, meso and macro, and summarises the learnings from the tables.

## 4. Discussion

This study brought together a body of work to identify practical solutions to improve the poorer breast cancer outcomes for Indigenous women in Australia. According to the evidence produced in this body of work, the two outstanding factors that lead to the success of research intending to benefit Indigenous peoples are:The participation of the community in research and being researchers of one’s own concerns.Incorporation of culturally safe approaches in research design.

As with research intended to benefit Aboriginal and Torres Strait Islander peoples, it is also true that health service delivery is more effective if it is culturally safe and community-led. Having the community involved directly in the design and governance of health service delivery is the best chance of improving outcomes. In Australia, Aboriginal Community Controlled Health Organisations (ACCHOs) have been identified as key to improving health outcomes within Aboriginal and Torres Strait Islander communities. Breast cancer screening takes place externally at ACCHOs; however, their direct involvement will ensure increased uptake of screening. If we are to continue to screen Indigenous women in locations external to the ACCHO, it is imperative that there are Indigenous staff present at the screening premises and that the non-Indigenous service providers are aware of the cultural needs of the Indigenous women.

Cultural safety is integral to overcoming access and participation barriers, and there are various techniques to increase cultural safety, for example, an explicit welcome to Indigenous women (e.g., in the form of a statement in the foyer, flags and local artworks), the presence of Indigenous staff, a Reconciliation Action Plan, a zero-tolerance approach to racism and mandated cultural safety training for all staff [13,17,21]. Promotion of screening is also a crucial issue, providing information and explanation as a means of demystifying the process. Initiatives that raise awareness and facilitate increased uptake of screening and culturally safe care must be community led. As a highly practical measure, the relocation of portable screening services (e.g., a van) to a location more easily accessible for Indigenous women will likely improve attendance rates.

It is clear that breast cancer policy needs to be written by and with Indigenous people and updated to meet the needs of Indigenous women. Presently, policy acknowledges the barriers for Indigenous women in accessing effective breast cancer care; however, acknowledgement is simply not enough to facilitate meaningful change. Policy must provide appropriate direction and solutions for Indigenous women. Breast cancer policy in Australia is written for the majority of women and assumes adherence to its direction. This ignores the evidence that minorities face more barriers and subsequently experience lower rates of adherence. Policy should be based on the adherence of the health service provider, not the consumer.

This study identified nine strategies to improve breast cancer outcomes for Indigenous women by looking at barriers and enablers on three different levels, micro (interpersonal), meso (systemic) and macro (policy). Using the CMO framework, this study displays the possibilities of employing these strategies (as shown in Table 1) and, conversely, of making no changes at all (as shown in Table 2). This builds on the theory that if changes are not made, then the present circumstances will remain or worsen; Indigenous women in Australia will remain more likely than non-Indigenous women to die from breast cancer. On the other hand, drawing on the evidence gathered from Indigenous women with breast cancer [33], peer-reviewed evidence [29,37], policy and a panel of experts [31], these nine strategies for change have the potential to significantly improve outcomes. Perhaps most importantly, these strategies are practical, realistic and not difficult to implement.

This study looks at three different contexts to approach the issue comprehensively. Health service provision in Australia is informed by policy at the macro level, but the interactions between patient and clinician are commonly far removed from policy; proper interrogation needs to happen at the more granular levels, as does the relationship between macro policy, clinician behaviour and the patient experience. Not only breast cancer policy but also health provision settings and patient–clinician interactions need updating. Without change at all levels, there is less chance of significant and sustained improvement.

The recommendations in this research cover various aspects of the screening, detection, treatment and follow-up processes for breast cancer care and acknowledge that many aspects need to change. Increasing the knowledge and familiarity with the disease and improving health information fluency around early detection for the Indigenous women, while at the same time making sure that the clinician and health services understand and respond to the needs of the Indigenous women. The presence of Indigenous health service providers has been proven to make a positive difference [21,38] but this is not possible for every patient. Mainstream healthcare providers must ensure their services are inclusive of and respond to the needs of Indigenous women. It is not acceptable to assume that services designed for white middle-class women will meet the needs of Indigenous women. In fact, we know that they do not. Considering this, increased awareness and understanding of non-Indigenous health service providers is imperative to allow for a more constructive exchange between patient and clinician. Ensuring that the health care settings are culturally safe and that there are protocols in place that meet the needs of Indigenous women who receive abnormal results or a diagnosis will improve participation and ultimately breast cancer outcomes [32,33]. Lastly, it is imperative that policy be co-created by those for whom it is intended and that any new iteration of breast cancer policy in Australia at the very least has Indigenous women as co-creators.

In order to change entire systems, the overarching principles contained within the health policy must be modified. But it is also possible to make changes at the service delivery level, with a view to improving cultural safety and understanding of the disease. Relatively minor modifications to health information and ways of interacting can make a big difference. This study is the first to comprehensively look at the international, national and local context of breast cancer in Indigenous women in order to create practical solutions that are generalisable at the service delivery level.

The limitations of this study included the small sample size in the systematic review, the small study size and the NSW-only nature of the qualitative study. Additionally, the policy panel did not include representation of government agencies (although they were invited to participate).

## 5. Conclusions

It is proven that increased breast cancer screening participation leads to lower mortality rates. It is also clear that there are concrete ways to implement safer breast cancer screening and treatment pathways for Indigenous women in Australia. Increasing cultural safety will lead to increased participation in early detection and treatment. We know that to achieve parity, there must be a commitment to change, both in breast cancer policy and practice.

It seems unacceptable to be passive. Indigenous women in Australia continue to die at significantly greater rates than non-Indigenous women and require a purposeful, policy-driven, culturally safe approach for effective change. It is time to bring the evidence together to improve breast cancer prevention and treatment services. The changes required by policymakers and services are relatively straight-forward. Indigenous women are saying loudly and clearly that they need more information, that they need easier access and that they need Indigenous staff and community advocates. And we must listen.

## Figures and Tables

**Figure 1 cancers-16-01736-f001:**
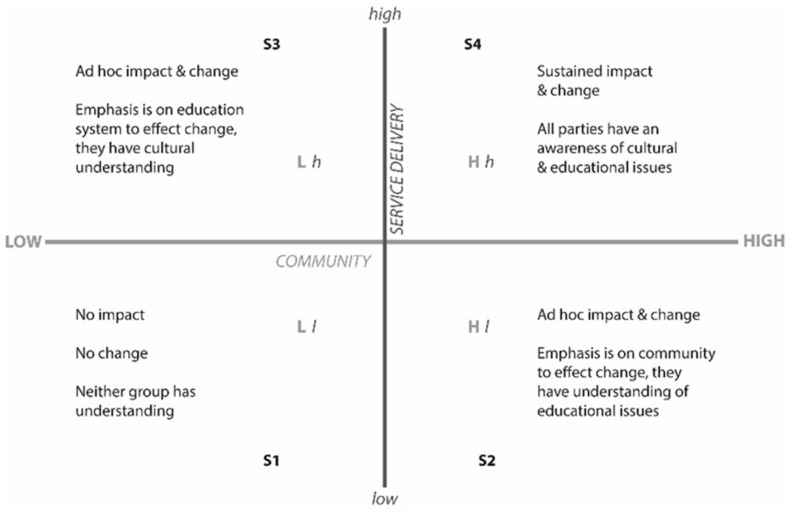
Communication, Consultation and Interaction, reproduced with permission from p. 32 of Community Led Research.

**Figure 2 cancers-16-01736-f002:**
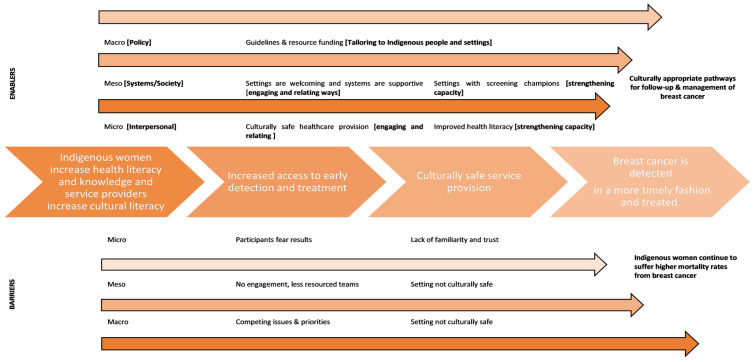
CMO framework when applied to breast cancer practice and policy in Australia.

**Table 1 cancers-16-01736-t001:** Representing the effect of changes to practice and policy.

Level	Context (If)	Outcome (Then)	Mechanism (Because)	Ultimate Outcome
Micro	Indigenous participants understand the importance of screening and early detection	More likely to attend screening	They are motivated to find out more and feel involved	1. Indigenous women attend routine breast cancer screening at the same/better rate than non-Indigenous women 2. Indigenous women are diagnosed with breast cancer earlier 3. Mortality rates from breast cancer for Indigenous women are comparable to those of non-Indigenous women in Australia
	Indigenous participants feel safe accessing screening services	More screening is likely to occur	Indigenous peoples find it culturally safe and welcoming	
	Providers/support staff are of similar background (Indigenous)	Indigenous clients are more likely to attend screening	Clients are more likely to understand and trust provider/support staff	
	Providers are local and familiar with Indigenous culture and community	Provider likely to encourage clients for screening	Providers are likely to understand local needs, customs and culture	
Meso	The screening setting is culturally safe	Clients are more likely to attend screening	Clients are more likely to get early diagnosis	
	Providers have screening champions in community	Clients are more likely to attend	Local champions convince their peers/fellow community members	
	There are culturally safe, tailored protocols for abnormal results catering to setting and community	Clients are more likely to access treatment	Screening results have a clearly outlined treatment/follow-up plan	
	There are support services available for people who receive diagnoses	Clients are more likely to adhere to treatment recommendations	Clients feel supported and culturally safe	
Macro	The policy is rewritten in consultation with Indigenous people and provides solutions to the acknowledged issues	Resources and funding will be provided to enact the policy, and service provision will be modified to be more culturally safe	Clients feel welcome and are more likely to engage with early detection and treatment services	

**Table 2 cancers-16-01736-t002:** Representing what will happen if changes to practice and policy are not made.

Level	Context (If)	Outcome (Then)	Mechanism (Because)	Ultimate Outcome
Micro	Indigenous clients do not understand the preventive and early intervention benefits of routine breast screening	More likely to not screen	Participants fear the implications of results	Low rates of screening and detection will continue, and mortality rates will not improve.
	Indigenous participants do not feel safe accessing screening services	Less likely to screen	Cultural safety improves attendance and adherence	
	Providers/support staff are not of a similar background (Indigenous)	Indigenous clients are less likely to attend screening	Indigenous clients are less likely to trust non-Indigenous staff	
	Providers are not local and not familiar with Indigenous culture and community	Indigenous clients do not feel comfortable and will avoid seeing their provider	Providers are less likely to understand local needs, customs and culture	
Meso	The screening setting is not culturally safe	Clients less likely to attend screening	Clients are less likely to get an early diagnosis	
	Providers do not have screening champions in community	Clients are less likely to attend	Local champions convince their peers/fellow community members	
	There are not culturally safe, tailored protocols for abnormal results catering to setting and community	Clients are less likely to access treatment	Clients are less likely to feel comfortable and welcome	
	There are no support services available for people who receive diagnoses	Clients are less likely to adhere to treatment recommendations	Clients are more likely to adhere when feeling supported	
Macro	Policy is not written by and in consultation with Indigenous people and does not change	Service provision remains culturally unsafe	Clients do not feel welcome and are not more likely to engage with early detection and treatment services	

## Data Availability

The original contributions presented in the study are included in the article, further inquiries can be directed to the corresponding author.

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
