# Peer review of "Improving Breast Cancer Outcomes for Indigenous Women in Australia"

_cancers, 2024, doi:10.3390/cancers16091736_

Round 1
Reviewer 1 Report
Comments and Suggestions for Authors
The theme of presented article are the plans and achievable solutions for improvement of breast cancer outcomes for Indigenous Women in Australia. The incidence rate of breast cancer is lower in Aboriginal and Torres Strait Islander classified as Indigenous women than non- Indigenous women in Australia. The worrisome fact is that mortality rate for Indigenous women is much higher.
The authors provided practical recommendations for adaptation of screening, treatment and follow up programs for breast cancer to improve screening rates, improve health outcomes and survival rates of the Indigenous women in Australia.
The theme of the presented article of great importance and addresses an important and timely topic, since there is a necessity for increasing awareness about breast cancer and providing accessibility to healthcare and screening especially in most deprived areas.
The article is well written, fluent and well organized. The language is clear and reader-friendly. The reference list covers the relevant literature with recently published articles.
I have no objections, in my opinion this study is suitable for publication in the current form.
Author Response
Thank you very much for your comments regarding the paper. We too believe it is important and appreciate the feedback.
Kind regards
Vita Christie
Reviewer 2 Report
Comments and Suggestions for Authors
Authors present a work addressing: ‘Improving Breast Cancer Outcomes for Indigenous Women in Australia’. This is an interesting study. However, the paper presents a few major issues including:
1. I believe that the abstract should be modified, there is no need to reveal the heart of the entire paper in the abstract, i.e nine practical ways to improve health outcomes and survival rates. Since authors repeat the same info in lines 149-164.
2. I would recommend the authors to re-organize all the tables and figures in order to increase readability.
3. Please provide subsections in the methodology section including, Study design, Patients/Protocol selection, Procedures and assessment, etc.
4. What is the novelty of this study? and how your outcomes can help further studies in this area. Please add a brief explanation of the novelty of this study in the introduction section.
5. Please also describe in the first paragraph of the results: Protocol description.
6. Please in discussion section add paragraph related to how we can applicate your results into practice?, why your work is valuable in the field? and how can we generalize the results?
Minor:
1. Please provide the limitations of the study.
Author Response
Thank you for the review of our paper. Please find below our responses and an updated manuscript attached.
|
Reviewer comment |
Authors’ response |
|
I believe that the abstract should be modified, there is no need to reveal the heart of the entire paper in the abstract, i.e nine practical ways to improve health outcomes and survival rates. Since authors repeat the same info in lines 149-164. |
Thank you for this feedback, we have modified in line with recommendations. |
|
I would recommend the authors to re-organize all the tables and figures in order to increase readability. |
Thanks for this recommendation. The journal have arranged the figures and tables according to guidelines so we will leave them as is. |
|
Please provide subsections in the methodology section including, Study design, Patients/Protocol selection, Procedures and assessment, etc. |
Thanks for these suggestion, we have edited the text to include. |
|
What is the novelty of this study? and how your outcomes can help further studies in this area. Please add a brief explanation of the novelty of this study in the introduction section. |
Thanks for pointing this out- we have added some text at the end of the Introduction section. |
|
Please also describe in the first paragraph of the results: Protocol description. |
Thanks for this- we have added to the first paragraph of the Results section. |
|
Please in discussion section add paragraph related to how we can applicate your results into practice?, why your work is valuable in the field? and how can we generalize the results? |
Thank you for this suggestion, we have added some text at the end of the Discussion section. |
|
Please provide the limitations of the study. |
We have added a Limitations section after the Discussion section. |
Round 2
Reviewer 2 Report
Comments and Suggestions for Authors
I recommend to accept the manuscriptin it’s present form.